# Parenchyma Sparing Anatomic Liver Resections (Bi- and Uni-Segmentectomies) for Liver Tumours in Children—A Single-Centre Experience

**DOI:** 10.3390/cancers16010038

**Published:** 2023-12-20

**Authors:** Maciej Murawski, Hanna Garnier, Joanna Stefanowicz, Katarzyna Sinacka, Ewa Izycka-Swieszewska, Malgorzata Sawicka-Zukowska, Pawel Wawrykow, Grazyna Wrobel, Agnieszka Mizia-Malarz, Patrycja Marciniak-Stepak, Piotr Czauderna

**Affiliations:** 1Department of Surgery and Urology for Children and Adolescents, Medical University of Gdansk, 80-210 Gdansk, Poland; garnier@gumed.edu.pl (H.G.); pczaud@gumed.edu.pl (P.C.); 2Department of Pediatrics, Hematology and Oncology, Medical University of Gdansk, 80-210 Gdansk, Poland; jstefanowicz@gumed.edu.pl; 3Department of Radiology, Medical University of Gdansk, 80-210 Gdansk, Poland; katarzynasinacka@op.pl; 4Department of Pathology and Neuropathology, Medical University of Gdansk, 80-210 Gdansk, Poland; eczis@gumed.edu.pl; 5Department of Pediatric Oncology and Hematology, Medical University of Bialystok, 15-089 Bialystok, Poland; malgorzata.sawicka-zukowska@umb.edu.pl; 6Department of Pediatric Oncology, Pomeranian Medical University, 71-210 Szczecin, Poland; p.wawrykow@spsk1.szn.pl; 7Department and Clinic of Haematology, Blood Neoplasms, and Bone Marrow Transplantation, Medical University of Wroclaw, 50-425 Wroclaw, Poland; wrobel.wroc@wp.pl; 8Department of Pediatric Oncology, Hematology and Chemotherapy, Medical University of Silesia, 40-752 Katowice, Poland; a.mizia@wp.pl; 9Department of Pediatric Oncology Hematology and Transplantology, Poznan University of Medical Sciences, 60-572 Poznan, Poland; pmarciniak@ump.edu.pl

**Keywords:** segmentectomy, liver, tumour, children

## Abstract

**Simple Summary:**

Liver tumours in children are definitely “surgical tumours”, and their complete resection is essential for cure. Compared to adults, the extent of resection in children can be much greater; up to 75–85% of the liver parenchyma can be safely resected. However, parenchymal preservation resection (segmentectomy, bisegmentectomy) seems to be a good option in carefully selected patients. There are very few publications regarding minor liver resections in children. From the available literature and data presented here, we propose that (bi)segmentectomy can become a viable surgical option in patients with favourable tumour locations and the absence of portal invasion in imaging. But, when qualifying a child for minor resection, it is important to remember that meeting oncological goals remains the primary objective of liver surgery. Further studies evaluating the impact of parenchymal preservation surgery on surgical and oncological outcome after liver resection in children should be conducted with a larger dataset.

**Abstract:**

*Purpose: To present* a single-centre experience in bi- and uni-segmentectomies for primary liver tumours in children. Methods: This study included 23 patients that underwent (bi)segmentectomy. There were 15 malignant tumours (hepatoblastoma—13 patients), 7 benign tumours, and 1 calcifying nested stromal epithelial tumour. Results: The median tumour diameter was 52 mm (range 15–170 mm). Bisegmentectomy 2–3 was most frequently performed (seven patients), followed by bisegmentectomy 5–6 (four patients). The median operative time was 225 min (range 95–643 min). Intraoperative complications occurred in two patients—small bowel perforation in one and an injury of the small peripheral bile duct resulting in biloma in the other. The median resection margin in patients with hepatoblastoma was 3 mm (range 1–15 mm). Microscopically negative margin status was achieved in 12 out of 13 patients. There were two recurrences. After a median follow-up time of 38 months (range 12–144 months), all 13 patients with HB were alive with no evidence of disease. Two relapsed patients were alive with no evidence of disease. Conclusions: From the available literature and data presented here, we propose that (bi)segmentectomy can become a viable surgical option in carefully selected paediatric patients and is sufficient to achieve a cure. Further studies evaluating the impact of parenchymal preservation surgery on surgical and oncological outcome should be conducted with a larger dataset.

## 1. Introduction

Surgery remains the cornerstone of the management of primary liver tumours, and achieving complete resection is essential for cure [1,2,3]. Hemihepatectomy *has been acknowledged as the standard procedure in children with liver tumours*. Due to the significant progress made in surgical armamentarium and operative techniques, complex liver resections in children with minimal operative morbidity and mortality have become possible [4]. On the other hand, parenchymal preservation surgery is feasible in carefully selected patients. *The aim of this study was to present* a single-centre experience in formal segmentectomies for primary liver tumours in children.

## 2. Materials and Methods

Between May 2011 and September 2022, 87 consecutive children with liver tumours underwent liver resection. Among these patients, 23 had (bi)segmentectomy (16 boys and 7 girls, aged between 4 months and 13 years). There were 15 malignant tumours (hepatoblastoma—13, sarcoma—1, nephroblastoma metastasis—1), 7 benign tumours (focal nodular hyperplasia, FNH—4, hamartoma—2, vascular lesion—1), and 1 calcifying nested stromal epithelial tumour (CSNET). CNSET is a very rare primary hepatic tumour with nonspecific clinical features and low malignant potential. Differential diagnoses include hepatic vascular tumours, fibrolamellar HCC, and hepatoblastoma [5]. Diagnoses and resection types of all patients with liver tumours operated on between 2011 and 2020 are presented in Table 1. Demographics and clinical data of patients undergoing segmentectomy are summarised in Table 2. We did not compare the analysed group with the group of patients who underwent other resections (mainly hemihepatectomy) since, in our opinion, these two groups of patients are incomparable because only patients with tumours of favourable location were eligible for (bi)segmentectomy. The second problem was the fact that a limited number of (bi)segmentectomies were available to compare with other resections.

## 3. Definitions

Segmentectomy is a complete resection of a part of the liver parenchyma supplied by a segmental branch of the portal vein [6]. The term segmentectomy in this publication is used to mean the anatomical resection of segments 1–8 based on Couinaud’s division of liver anatomy. Figure 1 and Figure 2 show intraoperative images of the patients suitable for bisegmentectomies.

## 4. The Selection Criteria for (Bi)Segmentectomy

The criteria for (bi)segmentectomy were generally based on preoperative radiological features and intraoperative evaluation. Our indications for (bi)segmentectomy were as follows: (1) favourable location of the tumour (tumour extension confined to one or two hepatic segments only), (2) no hepatic vein or inferior vena cava involvement, (3) no main portal vein or both first-order portal venous involvement, (4) the ability to obtain adequate macroscopic resection margin, and (5) the availability of a team experienced in liver surgery (high-volume centre). There are, however, several aspects worth paying attention to. First of all, good knowledge of liver anatomy of the portal vessels (with a high anatomical variability!) and high-quality imaging (Doppler US, CT and/or MRI) are necessary to plan parenchymal preservation resection. Secondly, this surgery relies on advanced intraoperative ultrasound (US-guided surgery); thus, a very experienced radiologist is needed. Intraoperative ultrasound was performed in all cases. And finally, the most important aspect was the favourable location of the tumour and not so much its size.

## 5. Results

The median tumour diameter was 52 mm (range 15–170 mm). In two cases, laparoscopic bisegmentectomy 2–3 was performed. In the remaining patients, an open procedure was performed. In one case (sarcoma), partial *diaphragm resection* was performed en bloc with the *liver tumour*. Bisegmentectomy 2–3 was most frequently performed (seven patients), followed by bisegmentectomy 5–6 (four patients). The median operative time was 225 min (range 95–643 min). The median time of hospitalisation was 9 days (range 4–14). The intraoperative and postoperative variables of patients undergoing segmentectomy are shown in Table 2.

Intraoperative complications occurred in two patients (2/23—8.7%). A 2-year-old boy with hamartoma of the liver underwent laparoscopic bisegmentectomy 2–3. The procedure was complicated by a small bowel perforation caused by the retrieval bag during the *extraction* of the specimen from the abdominal cavity. The perforation was directly repaired by interrupted sutures. The second patient was a 13-year-old boy with FNH who underwent bisegmentectomy 7–8. An injury of the small peripheral bile duct (to segments 5–6) occurred during the operation. The leak was directly repaired by interrupted Prolene 6–0 sutures. The development of an intra-abdominal biloma was observed postoperatively. It was successfully treated conservatively (drained in situ).

In the analysed group, the most common diagnosis was hepatoblastoma (13 patients). This group of patients was analysed separately, and they are presented in Table 3. The median resection margin achieved was 3 mm (range 1–15 mm). A microscopically negative margin status was achieved in 12 out of 13 patients. There were two recurrences. A brief description of these patients is provided below.

The first patient was a 4-year-old girl with a large tumour in the right lobe of the liver (MRI 103 × 77 × 140 mm). Computed tomography revealed a metastatic lesion in the left lung. *After biopsy, hepatoblastoma was diagnosed. The patient was treated according to PHITT protocol (group D—high-risk HB patients). After three blocks of chemotherapy (with a good response of the main tumour: MRI 47* × *39* × *52 mm), she* had chemotherapy-induced complete remission of lung lesions and underwent bisegmentectomy 6–7. The resection was microscopically complete. Ten months after the resection, MR imaging revealed a lesion 20 × 16 × 22 mm in segment 5 of the liver. AFP was within the normal range. Initially stable, the lesion enlarged after 8 months of follow-up (29 × 20 × 22 mm). A core needle biopsy was performed, which confirmed a hepatoblastoma recurrence. The patient was operated on and nonanatomical liver resection was performed. The tumour was tightly attached to the abdominal wall. A fragment of the abdominal wall and segment 5 of the liver were removed. In the pathological examination, the tumour was penetrating the fatty tissue covering the outer surface of the specimen and was present at its cut-off line. The resection margin from the liver parenchyma was 5 mm. After surgery, the stage assessment was performed. Chest CT revealed multiple pulmonary nodules (suspected lung metastases). The patient underwent second-line chemotherapy. She received three cycles of Adriamycin and Cisplatin. A partial response of lung metastases was achieved. The child underwent VATS (Video-Assisted Thoracic Surgery) metastasectomy with real-time visualisation under *ICG (indocyanine green)* fluorescence. Two lesions from the left lung and one nodule from the right lung were removed. All the removed lesions were necrotic. The patient received six cycles of carboplatin and etoposide following metastasectomy. The child finished the second-line treatment in August 2023. Abdominal MRI and chest CT confirmed complete remission of the disease, and the girl has remained free of disease since then.

The second patient was a 3-year-old boy. In April 2020, he was diagnosed with a mass in the right lobe of the liver measuring 94 × 90 × 93 mm, which was found to be penetrating into the right branch of the portal vein. The mass was classified as PRETEXT II P+. The patient received preoperative chemotherapy, resulting in a reduction in tumour size (28 × 27 × 12 mm) and a tumour thrombus withdrawal from the portal vein. Based on the tumour’s increased distance from the portal vein, bisegmentectomy 5–6 was performed. Macroscopically radical resection was achieved. However, microscopic examination revealed the presence of cancer cells in the cut line of the portal vein branch. Subsequent monitoring revealed increasing alpha-fetoprotein (AFP) levels, despite no evidence of disease on imaging. To address the disease progression and small left lobe, the patient was qualified for Associating Liver Partition and Portal Vein Ligation for Staged Hepatectomy (ALPPS), a two-stage right hemihepatectomy procedure (5 months after bisegmentectomy). Histopathology assessment during ALPPS demonstrated the focal presence of emboli or neoplastic plugs in selected sections and uncertain resection borders, underscoring the aggressiveness of the disease. Despite the ALPPS procedure, AFP levels continued to rise, prompting the consideration of novel therapies. The patient was subsequently qualified for Chimeric Antigen Receptor T-cell (CAR-T) therapy. CAR-T therapy represents an innovative approach to cancer treatment, utilizing genetically engineered T-cells to recognise and target specific tumour-associated antigens and induce a cell-mediated attack that leads to tumour cell death [7]. Following CAR-T therapy, the patient exhibited temporary remission, but the tumour eventually recurred in segment 1 of the liver. The subsequent surgical intervention (August 2023) consisted of a resection of the first liver segment (32 months after ALPPS). At present, the patient remains under observation, and his AFP levels have returned to close-to-normal ranges, indicative of a positive clinical response to the recent intervention.

After a median follow-up time of 38 months (range 12–144 months), all 13 patients with HB were alive with no evidence of disease. One patient was lost to follow-up after 19 months. Two relapsed patients, described above, are alive with no evidence of disease. The first patient remains alive 16 months after the surgery for relapse and 31 months after the primary surgery. The second patient has also survived after undergoing surgery for a second relapse in August 2023.

Liver tumours, especially hepatoblastoma, are definitely “surgical tumours”, and only their complete resection is curative [1,2,3]. Anatomic resections are generally recommended, and hemihepatectomy is a standard procedure in children. However, parenchyma preserving resection (segmentectomy, bisegmentectomy) seems to be a good option in selected patients. The question arises whether it is worth performing segmentectomy considering the fact that it is more difficult and requires experience in liver surgery. Even though we are the reference centre for paediatric liver non-transplant surgery in Poland, “only” 23 (bi)segmentectomies during the last 11 years were performed. To the best of our knowledge, there are only three similar publications regarding minor liver resections in children, written by Qureshi et al. [8], Li et al. [4], and Liu et al. [9].

Traditionally, non-anatomic resections were believed to be associated with worse outcome [10], and hence they are rarely performed, typically in cases of multifocal tumours, when LTX is not viable due to metastatic disease. Qureshi et al. conducted a study involving 25 non-anatomic liver resections, comparing the results with 95 anatomic resections [8]. Their findings concluded that non-anatomic liver resection is feasible in carefully selected patients, with no positive margins observed. They noted similar rates of complications and outcomes between the two approaches. However, further studies are necessary to challenge established practices in managing primary liver tumours, particularly in hepatoblastoma cases.

A comprehensive understanding of liver anatomy and substantial experience in liver surgery are indispensable in determining the appropriate extent of liver resection. Claude Couinaud’s seminal report in 1954 delineating segmental liver anatomy significantly contributed to the reduction in surgical morbidity [11]. Subsequently, anatomic resections based on the division of liver anatomy as described (such as segmentectomy or hemihepatectomy) are generally recommended [12,13,14].

Segmentectomy is a procedure that Makuuchi et al. used for hepatocellular carcinoma (HCC) and reported in 1985 [15]. It is one of the standard operations for adult HCC [16]. Many patients with HCC have liver cirrhosis and chronic hepatitis. For this reason, extensive resections should be avoided to preserve liver function [6,16]. Liver preservation surgery also leads to a reduction in remnant liver ischemia and thus reduces the risk of biliary stasis or leakage [16,17,18].

In children with liver tumours, the situation is different. We usually operate on a “healthy liver”, so the extent of resection can be much greater; up to 75–85% of the liver parenchyma can be safely resected [19,20]. Despite this, in carefully selected cases, it is worth considering a smaller scope of resection, bearing in mind that the child still has their whole life ahead of them. However, when qualifying a child for (bi)segmentectomy, it is important to remember that meeting oncological goals remains the primary objective of liver surgery. For example, the Children’s Oncology Group (COG) guidelines recommend a segmentectomy, sectionectomy, or lobectomy for PRETEXT I and II tumours provided; however, a 1 cm resection margin should be maintained [3].

However, in light of recent studies, the necessity of adhering to the traditionally recommended tissue margin in paediatric HB cases is subject to debate. On the one hand, attaining a 1 cm margin of normal liver parenchyma can be challenging, particularly in young children, and might not be deemed necessary. On the other hand, we do not know the significance of the microscopically positive margin for prognosis. In 2019, Aronson et al. conducted an analysis involving children who were participants in the SIOPEL-2 and -3 studies. They specifically compared the outcomes of 58 patients with microscopically positive margins to 371 completely resected patients [21]. In both risk categories—SR and HR—no differences in the local recurrent rate, EFS, nor OS were observed between the two groups. So, in children operated on for hepatoblastoma, it could possibly be said that any margin is sufficient. Moreover, the identification of a positive margin on the specimen side does not automatically indicate the existence of tumour cells on the patient side. The tools employed for parenchymal dissection and haemostasis might contribute to eradicating tumour cells from the surface of the remaining liver. Additionally, the ongoing administration of chemotherapy after surgery could potentially eliminate micro-residuals. It is important to note that this formal analysis represents an initial and singular investigation regarding the influence of microscopically positive resection on the outcome of hepatoblastoma patients. The findings presented in this study necessitate further validation and confirmation through additional research. An answer to the above question may be given by a new PHITT study. In this context, it is extremely important to properly select patients with malignant liver tumours for liver preservation surgery. *The planned operation must be radical*, even at the cost of a major liver resection. That is why the pre- and intraoperative assessment of tumours is so important. Intraoperative ultrasound should be available in all cases. We need to know what the status of the vascular inflow and outflow is to the planned liver remnant [22].

The importance of proper patient selection is shown by both cases of local recurrence described above. The first patient had a negative resection margin after primary surgery but belonged to the high-risk group. It seems that the second patient was incorrectly qualified for bisegmentectomy 5–6 because his tumour initially penetrated into the right portal vein, albeit this portal involvement was resolved with preoperative chemotherapy. It seems that patients with high-risk tumours and/or initial portal vein involvement (P—positive tumours) should be scheduled for hemihepatectomy rather than segmentectomy. Of course, it is rather difficult to draw final conclusions based on 23 segmentectomies and 2 recurrences.

With regard to parenchymal preservation resection, the question arises whether this type of surgery is associated with a higher risk of complications. Li et al. reported 87 patients with liver tumours (44—malignant, 33—benign) [4]. The patients were divided into two groups depending on the type of surgery: minor (1 or 2 segments) and major liver resection (3 or more hepatic segments). There were 36 minor liver resections and 51 major liver resections.

This study indicated that the complication rate was comparable in both groups. A different observation was made by Liu et al. [9]. These authors reported 156 children undergoing liver resection. There were 82 benign lesions and 74 malignant tumours. Twenty-seven patients (27/156, 17.3%) underwent segmental resection, and in 21 cases, bisegmentectomy was performed (21/156, 13.5%). According to these conclusions, the number of hepatic segments removed had a significant impact on perioperative complications, with 14.5% for patients who underwent minor liver resection and 72.3% in patients who underwent the removal of over three segments. In our material, complications occurred in two patients (2/23—8.7%)—small bowel perforation caused by the retrieval bag in one case and biloma in another patient.

## 6. Conclusions

Further studies, preferably prospectively planned, evaluating the impact of parenchymal preservation surgery on surgical and oncological outcomes after liver resection in children should be undertaken with a larger dataset. From the available literature and data presented here, we propose that (bi)segmentectomy can become a viable surgical option in carefully selected paediatric patients with favourable tumour locations and is sufficient to achieve a cure. However, further multicentre prospective research is warranted to determine whether (bi)segmentectomy is a sufficient extent of resection. After performing 23 (bi)segmentectomies, the authors believe that it can be safely applied in children with liver tumours. Necessary conditions for performing segmentectomy are (1) favourable location of the tumour (away from the portal structures and major hepatic veins), (2) very good knowledge of liver anatomy, (3) the absence of portal invasion in imaging, (4) and experience in liver surgery (we are a high-volume centre for liver surgery in Poland). The most important thing, however, is to remember about oncological principles, even at the cost of a greater extent of resection.

## Figures and Tables

**Figure 1 cancers-16-00038-f001:**
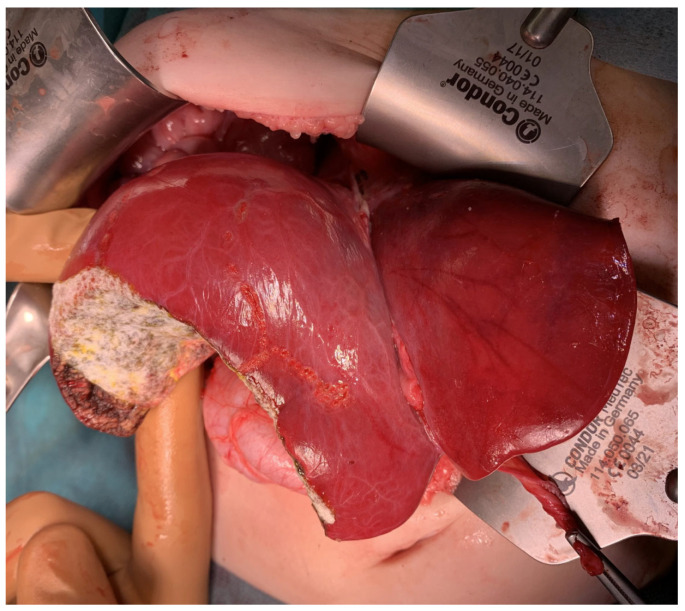
15-month-old boy with hepatoblastoma. Intraoperative photograph showing liver after segmentectomy 5 + 6.

**Figure 2 cancers-16-00038-f002:**
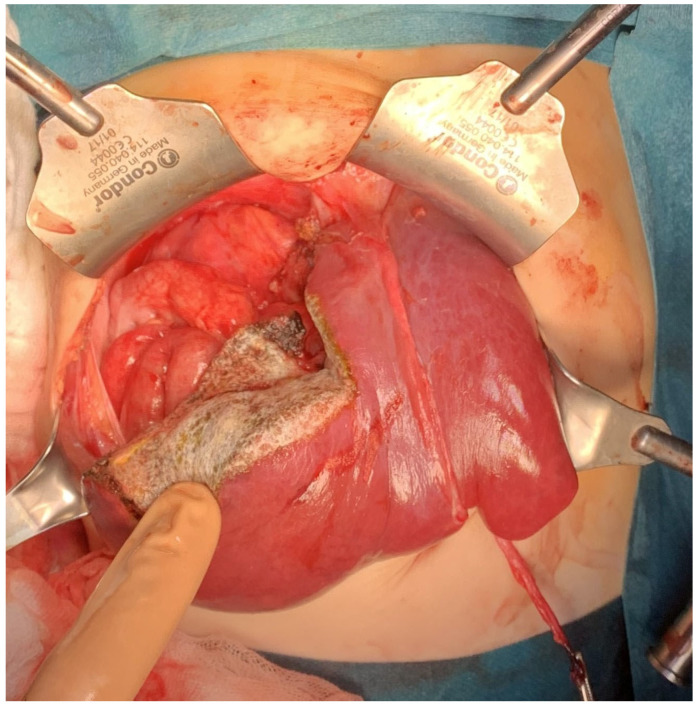
22-month-old boy with hepatoblastoma. Intraoperative photograph showing liver after segmentectomy 7 + 8.

**Table 1 cancers-16-00038-t001:** Diagnoses and resection types of all patients with liver tumours operated on between 2011 and 2020.

Tumour	Diagnoses	*n* = 87	Type of Resection	*n*
Malignant	Hepatoblastoma	61	Right hemihepatectomy	18
			Extended right hemihepatectomy	9
			Left hemihepatectomy	14
			Extended left hemihepatectomy	5
			ALPPS	2
			Segmentectomy	13
	HCC	4	Right hemihepatectomy	3
			Left hemihepatectomy	1
	Sarcoma	3	Right hemihepatectomy	1
			Extended right hemihepatectomy	1
			Segmentectomy	1
	Metastases	2	Nonanatomical resection	1
			Segmentectomy	1
	Rhabdoid tumour	1	Nonanatomical resection	1
Benign	FNH	6	Extended right hemihepatectomy	1
			Nonanatomical resection	1
			Segmentectomy	4
	Hamartoma	6	Right hemihepatectomy	1
			Extended right hemihepatectomy	3
			Segmentectomy	2
	Vascular tumour	2	Nonanatomical resection	1
			Segmentectomy	1
	Adenoma	1	Left hemihepatectomy	1
Other	CNSET	1	Segmentectomy	1

HCC, hepatocellular carcinoma; FNH, focal nodular hyperplasia; CNSET, calcifying nested stromal epithelial tumour; ALPPS, associating *liver* partition and portal vein ligation for staged hepatectomy.

**Table 2 cancers-16-00038-t002:** Demographics and intraoperative and postoperative variables of patients undergoing segmentectomy.

Demographics		
Age in months (median)	4–158 (20)	
Gender	female—7, male—16	
Diagnosis	Hepatoblastoma	13
	FNH	4
	Hamartoma	2
	UESL	1
	Vascular tumour	1
	CNSET	1
	Nephroblastoma metastasis	1
The largest tumour diameter in millimetres (median)	15–170 (52)	
Metastases	1—lung	
Intra- and postoperative variables		
Segment resected	Segment	*n*
	1	1
	2 + 3 *	7 *
	3	2
	5	2
	5 + 6	4
	6	1
	6 + 7	2
	7 **	1 **
	7 + 8	3
Duration of surgery in minutes (median)	95–643 (225)	
Intraoperative complications	1—intestinal perforation, 1—peripheral bile duct injury	
Hospital stay in days (median)	4–14 (9)	

FNH, focal nodular hyperplasia; UESL, undifferentiated embryonal sarcomas of the liver; CNSET, calcifying nested stromal epithelial tumour. * In 2 cases, laparoscopic bisegmentectomy was performed; in 1 case (sarcoma), the partial *diaphragm resection* was performed en bloc with the *liver tumour*. ** Segmentectomy 7 with simultaneous diaphragm resection was performed.

**Table 3 cancers-16-00038-t003:** Subgroup analysis of hepatoblastoma patients (*n* = 13).

Segment resected	Segment	*n*
	1	1
	2 + 3	2
	3	1
	5	2
	5 + 6	3
	6 + 7	2
	7 + 8	2
The largest tumour diameter in millimetres (median)	16–78 (50)	
Duration of surgery in minutes (median)	95–360 (240)	
Margin in millimetres	1–15 (3)	
Margin positive	1	
Recurrence	2	
Follow-up	38 months (range 12–144 months)	
Complete remission	13 (100%)	

## Data Availability

Data are contained within the article.

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
