# Peer review of "Parenchyma Sparing Anatomic Liver Resections (Bi- and Uni-Segmentectomies) for Liver Tumours in Children—A Single-Centre Experience"

_cancers, 2023, doi:10.3390/cancers16010038_

Round 1
Reviewer 1 Report
Comments and Suggestions for Authors
This study examined the feasibility of parenchyma sparing anatomic liver resections in children. The authors concluded that bi- and mono-segmentectomies can become a viable surgical option in carefully selected pediatric patients. This study may provide some useful information for pediatric surgeons but there are some drawbacks.
1. In this study, 23 segmentectomies were performed, 2 patients experienced surgical complications: small bowel perforation and postoperative biloma. This study consisted of 87 liver resections including hemihepatectomies and more than hemihepatectomies of 59 cases. To confirm the feasibility of segmentectomies, the authors should compare the surgical outcomes with hemihepatectomy group: surgical results such as bleeding and operative time, surgical complications, and long-term outcomes.
2. Two surgical resections were performed laparoscopically. Other liver resections were non-laparoscopically performed? Laparoscopic liver resections has impact on short and long-term outcomes.
Author Response
This study examined the feasibility of parenchyma sparing anatomic liver resections in children. The authors concluded that bi- and mono-segmentectomies can become a viable surgical option in carefully selected pediatric patients. This study may provide some useful information for pediatric surgeons but there are some drawbacks.
- In this study, 23 segmentectomies were performed, 2 patients experienced surgical complications: small bowel perforation and postoperative biloma. This study consisted of 87 liver resections including hemihepatectomies and more than hemihepatectomies of 59 cases. To confirm the feasibility of segmentectomies, the authors should compare the surgical outcomes with hemihepatectomy group: surgical results such as bleeding and operative time, surgical complications, and long-term outcomes.
Thank you very much for the review and for your valuable comment. As we described in the “Material and Methods” section, we did not compare the analyzed group with the hemihepatectomy group. In our opinion, these two groups of patients are incomparable, because only patients with favorable tumour location were eligible for (bi)segmentectomy (selection bias!). The main goal of our publication was to answer the question whether it is possible to avoid major liver resection in children with primary liver tumour and therefore preserve functional nontumour bearing arenchyma. Another problem with such a comparison is limited number of (bi)segmentectomies compared with other resections. Despite the modest numbers and retraspective nature of the study, the results of our analysis pleads not to abandon the minor resections (segmentectomies) completely. We need further studies evaluating the impact of parenchymal preservation surgery on surgical and oncological outcome after liver resection in children. These studies should be taken on a larger data set. Perhaps in the future, a larger number of patients will allow us to make such a comparison.
- Two surgical resections were performed laparoscopically. Other liver resections were non-laparoscopically performed? Laparoscopic liver resections has impact on short and long-term outcomes.
That's right, we performed laparoscopic resection in 2 cases. In the remaining patients, we performed an open procedure. The “Results” section was updated according to your suggestion.
There are very few reports regarding laparoscopic liver resections in children in an english literature. In 2021, we published an article presenting our experiences with laparoscopic liver resection [1]. We reported 6 cases of laparoscopic hepatic resection and demonstrated the feasibility of this method in children with small benign tumors. Its role in malignant liver tumors remains controversial, particularly with regard to oncological safety. As with segmentectomy, laparoscopic liver resection can only be performed in carefully selected patients.
[1] Murawski M, Losin M, Golebiewski A et al. Laparoscopic resection of liver tumors in children. J Pediatr Surg 2021 Feb;56(2):420-423.
Reviewer 2 Report
Comments and Suggestions for Authors
The study by Murawski et al. evaluated the use of segmentectomy for primary liver tumors in children. Out of 23 patients, 15 had malignant tumors, mostly hepatoblastoma. The most common procedure was bi-segmentectomy 2-3. Despite two complications, the study suggests that segmentectomy is a viable option for selected pediatric patients, with good outcomes observed, including a median follow-up of 50 months where all hepatoblastoma patients remained alive with no evidence of disease. With the inherent challenges of small sample sizes in pediatric populations, a study involving 23 patients is adequate. The manuscript provides a succinct summary highlighting the often overlooked role of segmentectomy in pediatric cases. It offers useful insights into the selection of surgical methods for children with liver tumors. Some revisions are suggested.
1. While the authors have provided a clear explanation of the criteria for selecting segmentectomy, it would enhance the manuscript to incorporate visual evidence such as imaging and intraoperative findings. This addition would not only strengthen the comprehensibility of the selection process but also offer a more comprehensive and tangible understanding for readers. Including visuals could aid in illustrating the practical application of the outlined criteria, thereby enhancing the overall clarity and impact of the manuscript.
2. The authors appear to have employed a 3 mm resection margin, which is notably shorter than values reported in other publications. It would be beneficial for the manuscript to include a rationale behind this specific choice.
3. It is ideal to provide the information about the number of patients who underwent laparoscopic segmentectomy and details on the outcomes compared with open surgeries.
4. The authors have explained the rationale for not analyzing the comparison between hemihepatectomy and segmentectomy in general patients. However, it is suggested that the manuscript could gain valuable insights by specifically exploring this comparison within the subset of patients with hepatoblastoma, given that this group constitutes the largest population in the authors' center.
5. Moderate English editing is necessary.
Comments on the Quality of English Language
Moderate editing.
Author Response
Review 2
The study by Murawski et al. evaluated the use of segmentectomy for primary liver tumors in children. Out of 23 patients, 15 had malignant tumors, mostly hepatoblastoma. The most common procedure was bi-segmentectomy 2-3. Despite two complications, the study suggests that segmentectomy is a viable option for selected pediatric patients, with good outcomes observed, including a median follow-up of 50 months where all hepatoblastoma patients remained alive with no evidence of disease. With the inherent challenges of small sample sizes in pediatric populations, a study involving 23 patients is adequate. The manuscript provides a succinct summary highlighting the often overlooked role of segmentectomy in pediatric cases. It offers useful insights into the selection of surgical methods for children with liver tumors. Some revisions are suggested.
Thank you for your favorable assessment of our effort.
- While the authors have provided a clear explanation of the criteria for selecting segmentectomy, it would enhance the manuscript to incorporate visual evidence such as imaging and intraoperative findings. This addition would not only strengthen the comprehensibility of the selection process but also offer a more comprehensive and tangible understanding for readers. Including visuals could aid in illustrating the practical application of the outlined criteria, thereby enhancing the overall clarity and impact of the manuscript.
Thank you very much for the review and for your valuable comments. We have added intraoperative imaging of two patients
Fig. 1 15-month-old boy with hepatoblastoma. Intraoperative photograph showing liver after segmentectomy 5+6.
Fig. 2 22-month-old boy with hepatoblastoma. Intraoperative photograph showing liver after segmentectomy 7+8.
- The authors appear to have employed a 3 mm resection margin, which is notably shorter than values reported in other publications. It would be beneficial for the manuscript to include a rationale behind this specific choice.
Thank you for the comment. It's not exactly like that. In the “Results" section and in Table 3 we presented that in analyzed group the median resection margin was 3 mm (range 1-15 mm). It is obvious that the goal of surgical resection is to achieve complete tumor clearance, but advocated 1-cm margin in pediatric HB is a matter of debate and remains rather controversial nowadays. On the one hand, 1-cm margin of normal liver parenchyma is sometimes difficult to achieve especially in young children and probably not required. There are several papers which document that even minimal margin in hepatoblastoma, which is the most common pediatric liver malignant neoplasm, is sufficient. On the other hand, we do not know the significance of the microscopically positive margin for prognosis. Based on Aronson's publication [1], one may wonder whether any margin will be sufficient in children operated on for hepatoblastoma. Nevertheless in our center, we always try to achieve the largest possible resection margin and opting for liver parenchyma sparing anatomic resections we have always kept in mind the need for obtaining adequate surgical margin
- It is ideal to provide the information about the number of patients who underwent laparoscopic segmentectomy and details on the outcomes compared with open surgeries.
We performed laparoscopic resection in 2 cases. In the remaining patients, we performed an open procedure. Because of the modest numbers we did not compare the two patients (one with hamartoma, the second with CNSET) who underwent laparoscopic segmentectomy with the rest of segmentectomy group. Laparoscopic liver resections in children are a topic for separate researches.
There are very few reports regarding laparoscopic liver resections in children in an english literature. In 2021, we published an article presenting our experiences with laparoscopic liver resection [1]. We reported 6 cases of laparoscopic hepatic resection and demonstrated the feasibility of this method in children with small benign tumors. Its role in malignant liver tumors remains controversial, particularly with regard to oncological safety.
[1] Murawski M, Losin M, Golebiewski A et al. Laparoscopic resection of liver tumors in children. J Pediatr Surg 2021 Feb;56(2):420-423.
- The authors have explained the rationale for not analyzing the comparison between hemihepatectomy and segmentectomy in general patients. However, it is suggested that the manuscript could gain valuable insights by specifically exploring this comparison within the subset of patients with hepatoblastoma, given that this group constitutes the largest population in the authors' center.
Thank you very much for this valuable comment. As we described in the “Material and Methods” section, we did not compare the analyzed group with the hemihepatectomy group. In our opinion, these two groups of patients are incomparable, because only patients with favorable tumour location were eligible for (bi)segmentectomy (selection bias!). The main goal of our publication was to answer the question whether it is possible to avoid major liver resection in children with primary liver tumour and therefore preserve functional non-tumour bearing parenchyma. Another problem with such a comparison is limited number of (bi)segmentectomies compared with other resections. Despite the modest numbers and retrospective nature of the study, the results of our analysis plead not to abandon the minor resections (segmentectomies) completely. We need further studies evaluating the impact of parenchymal preservation surgery on surgical and oncological outcome after liver resection in children. These studies should be taken on a larger data set. Perhaps in the future, a larger number of patients will allow us to make such a comparison.
- Moderate English editing is necessary.
Thank you again for your review and valuable comments. We have re-edited paper’s English.
Round 2
Reviewer 1 Report
Comments and Suggestions for Authors
I think that this manuscript is acceptable.
Reviewer 2 Report
Comments and Suggestions for Authors
My questions have been addressed.